# Power-Based Side-Channel Attacks on Program Control Flow with Machine Learning Models

Andey Robins [1,*], Stone Olguin [2], Jarek Brown [2], Clay Carper [2] and Mike Borowczak [1]

1 Department of Electrical and Computer Engineering, University of Central Florida, Orlando, FL 32816, USA; mike.borowczak@ucf.edu
2 Department of Electrical Engineering and Computer Science, University of Wyoming, Laramie, WY 82070, USA; aolguin1@uwyo.edu (S.O.); jbrow125@uwyo.edu (J.B.); ccarper2@uwyo.edu (C.C.)
* Correspondence: ja548335@ucf.edu

**Abstract:** The control flow of a program represents valuable and sensitive information; in embedded systems, this information can take on even greater value as the resources, control flow, and execution of the system have more constraints and functional implications than modern desktop environments. Early works have demonstrated the possibility of recovering such control flow through power-based side-channel attacks in tightly constrained environments; however, they relied on meaningful differences in computational states or data dependency to distinguish between states in a state machine. This work applies more advanced machine learning techniques to state machines which perform identical operations in all branches of control flow. Complete control flow is recovered with 99% accuracy even in situations where 97% of work is outside of the control flow structures. This work demonstrates the efficacy of these approaches for recovering control flow information; continues developing available knowledge about power-based attacks on program control flow; and examines the applicability of multiple standard machine learning models to the problem of classification over power-based side-channel information.

**Keywords:** side-channel attack; machine learning; power analysis; cybersecurity; control flow; dynamic program analysis

## 1. Introduction

A finite-state machine (FSM) is a computation model commonly used within the embedded system space; program control flow in embedded devices is often handled by an FSM. Smaller state machines, those with fewer states and transitions, can exist in limited purpose devices, such as vending machines, or more complex devices such as telecommunications devices. Such devices are often the target of Side-Channel Analysis (SCA), which aims to recover information from an embedded device. Power-based side-channel data are most commonly measured directly from the device via an instrumented VCC line. While this method requires direct access to the victim device, it is the most common method for gathering data for SCA. Other common side-channels utilized in SCA include system byproducts such as electromagnetic radiation, sound, and heat.

With a constant increase in consumer usage of Internet of Things (IoT) devices, low-power embedded systems are constantly being put into operation. One popular example of such a system making use of an FSM are smart locks. Often, these devices allow an end-user to configure multiple access codes and enable logging when a code is used. Access may be controlled remotely, allowing for on-the-fly adjustments to user access permissions. With the growing popularity of smart locks and similar smart devices, if an attacker were able to reverse-engineer the control flow inherent to the device, they may be able to influence behavior within the FSM. One informational prerequisite to building out such an attack would likely include identifying and characterizing the FSM responsible for whether or not the lock is engaged. Further, being able to exfiltrate sensitive data, such as

the secret key that determines the lock's activation state, could allow an attacker to bypass the smart lock, granting them unimpeded access to the end-user's home.

This work extends the previous work of Carper et al. [1], which performed Differential Power Analysis (DPA) on data collected through the use of the ChipWhisperer hardware platform [2]. Specifically, the ChipWhisperer Nano was used to gather power-trace data. An FSM that consisted of two states conducting identical operations, in conjunction with an oracle-based input guiding program control flow, was utilized for data collection. The resulting power-traces were used to train multiple machine learning classification algorithms. These trained algorithms were then used to differentiate between state transitions occurring during code execution on the microcontroller. Further, recovery was successful when applied to 256 distinct classes of state transitions, resulting in the identification of the underlying process control flow. This work deviates from the prior foundational work by exploring additional machine learning backed solutions to the problem of control flow recovery, as well as exploring homogeneous operations within an individual state's computations to experimentally determine what degree of divergence in state behavior is necessary to ensure program control flow recovery.

Contributions of this work seek to answer three research questions. First, is different behavior required in each state of an embedded state machine in order to completely recover the transitions? Second, to what degree does the proportion of time spent in control flow and in a particular state of the FSM impact the recoverability of the state transition ordering? Finally, how effective are "off-the-shelf," meaning algorithms with no manual configuration, machine learning models when applied to the task of recovering control flow information?

This work is organized as follows. Section 2 motivates the side-channel analysis space and establishes relevant background information. Section 3 outlines the data collection and classification process undertaken in this work. Section 4 outlines the classification accuracy and Section 5 examines the application of results to the research questions. Section 6 concludes this work with an acknowledgment of limitations binding this work and commentary on future research directions required to better understand the control flow recovery space using power-based side-channels.

## 2. Related Works

The history of power-based side-channel attacks has had a number of meaningful advances coming in the last few decades. Despite the coming exploration into the efficacy of such power-based attacks, recovery of higher-order, control-level information such as the execution path through a state machine was under-explored until very recently. Thus, this section presents a brief overview of relevant findings in power-based side-channels, along with its applications and crossover with the field of Automated Machine Learning (AutoML), and characterizes the relevant previous work to motivate the experiments conducted herein. AutoML, the practice of automatically searching machine learning pipelines for effective ML configurations, has begun to be applied to other side-channel problems such as cryptographic key recovery with high success, motivating attempts to apply AutoML findings to the problem of program control flow recovery.

### 2.1. SCA Backgrounds

Common side-channel attacks involve attacks on the power usage of a device. The most common of these are simple power analysis (SPA) and differential power analysis (DPA) [3]. These methods of side-channel analysis inspired research in device-level SCA; they have even been referred to as the "bedrock" for SCA research [4]. The direct analysis of power usage by gathering power traces allows for a user to be able to understand the implementations of cryptographic operations of a device, and it can allow for the extraction of a secret key. SPA and DPA are both considered passive attacks as they entail only observing various properties of the device; however, active attacks are another avenue explored in the literature. Fault injection attacks, a breadth of attacks which span

voltage glitching to temperature extremes, are another frequently explored avenue for SCA [5,6]. Other powerful techniques exist, such as combining different passive and active attack strategies. For example, a novel analysis tool Differential Behavioral Analysis (DBA) is a combination of a Safe Error Attack (SEA) and DPA [7]. DBA can be defended against using traditional bit-masking strategies. This work makes use of passive attacks exclusively, and specifically employs simple power analysis as a means to recover control flow information.

One of the most common applications of side-channel attacks is in the subversion of cryptographic systems. These attacks on crypto-schema employ both active and passive attacks. The injection of faults has been shown to be able to effectively recover a secret AES key, and the power observation making use of DPA similarly is able to recover a secret AES key [8,9]. Many of the early works in SCA demonstrated various ways to recover secret cryptographic information, either keys or text, to some degree; however, with the advent of post-quantum cryptography, many of the crypto-systems currently employed will become insecure and obsolete [10]. For more discussion of the topic, we refer the curious reader to one of the surveys on the state-of-the-art in the field [11,12]. The realm of side-channels is evolving in cooperation with these searches for quantum-resistant algorithms, and early works have demonstrated strong recovery attacks for cryptographic secrets in post-quantum, lattice-based cryptographic systems [13].

The rising prevalence of SCA in cryptographically sensitive applications and its impact to the security of a device led to development of different countermeasures against SCA. Borowczak and Vemuri developed a method to create side-channel resistant finite-state machines (S*FSMs) [14]. This introduced an algorithm to transform FSMs to side-channel resistant FSMs. The method of Random Process Interrupts (RPIs) allows for some of the operations in cryptographic devices to be less vulnerable to timing attacks by implementing strategic delays in execution [15]. In addition to the RPI method, two other popular methods of implementing countermeasures against side-channel power include masking and hiding [16]. Masking involves generating a random "mask" value that will attempt to conceal any intermediate value during cryptographic operations. Masking removes correlations between the gathered traces and the cryptographic secret information. Hiding involves trying to make traces appear to be random. This randomness can appear by adding noise to the power or implementing random delays or desynchronization. As a result, the gathered traces are harder to extract secret information. Therefore, many countermeasures against SCA have been devised, but DPA methods would still find secure information even with countermeasures in place. In the same work that introduced RPIs, the countermeasure is still shown to be vulnerable to DPA [15]. Thus, even with countermeasures in place, improvements of applications of SCA can circumvent these protections and secret information can still be extracted.

Instead of collecting and analyzing arbitrary power traces to determine the leakage of information Test Vector Leakage Assessment (TVLA) can be used [17]. TVLA involves using statistical tests to determine if there is significant evidence to determine if the device had any leakage. Goodwill et al. utilized the Welch's $t$-test to determine if there is a significant difference between two groups. The hypothesis tested whether the gathered traces are truly random or if there is leakage present within traces. While this methodology can be beneficial and has been used recently to show side-channel vulnerabilities in some of the NIST lightweight cryptography round 2 candidate s-boxes [18,19], TVLA can have issues demonstrated by relatively high rates of false negatives or false positives, even when using different tests such as the Pearson $\chi^2$ test [20].

Signal Processing also can be used for SCA both as an attack vector and for defensive countermeasures, as mentioned by Le et al. [21]. In addition, Le et al. also demonstrates how three different signal processing techniques could be applied to SCA, which would allow for more ways to implement it. The work also defends using signal processing to mitigate SCA. There is also a precedent for AutoML being utilized for signal processing; an example uses the AutoML procedure of acquiring correct hyperparameters for deep

learning to classify Electroencephalography (EEG) signals [22]. The work also claims that EEG signal classification is complex enough that machine learning techniques such as deep learning are "appropriate to find the best solutions". The hyperparameter tuning and deep learning from AutoML were stated to have less overfitting, and thus, yielded more optimized models for classifying the EEG signals.

Time-series forecasting, although prominently implemented with traditional machine learning methods, has significantly less implementation within AutoML toolchains, being cited as "still in the development stage" [23]. It is demonstrated in their review article that there are gaps in traditional time-series forecasting with machine learning in terms of reapplying AutoML to time-series methods that used traditional machine learning [23]. The article showcases that there are research avenues, such as deep learning or neural architecture search (NAS), to implement AutoML for time-series analysis.

AutoML has been implemented in analyzing time-series data, as experimented previously in comparison to rigorous, hand-crafted machine learning models [24]. In this paper, its authors conclude that in short-term models for time-series predictions, AutoML does not outperform traditional methods of using Machine Learning by manually tuning hyperparameters and preprocessing data. However, suggestions for time-series analysis with AutoML are: validation on the selection strategy with statistical significance tests; adding permutation strategies; and considering the cost of using AutoML with the benefit of its implementation. Additional implementation of AutoML utilize the process of NAS to efficiently search for an effective neural network architecture for utilization on time-series data [25]. It mentions that AutoML significantly improved the performance of searching a data-augmented time-series neural network architecture. The significance of NAS's performance boost was such that it outperformed other "best" statistical models. In terms of future possible improvements, the authors mention that further performance enhancements on the data augmentation could come from using other deep learning models such as GRU-AE and ConvLSTM.

One of the implementations of machine learning for side-channel analysis is named Deep Learning-based Side-Channel Analysis (DL-SCA) [26]. DL-SCA is a new area of research, which is signified by a large increase of papers on this topic. An advantage of DL-SCA includes more powerful analysis by taking up to a factor of five times less data to break through targets with countermeasures as compared to template attacks. DL-SCA also requires little to no effort when it comes to preprocessing and preparing the attack of the side-channel measurements. Related to this deep learning AutoML approach of side-channel analysis is the Deep Learning Leakage Assessment (DL-LA), a method of verifying that a trace has significant leakage information [27]. DL-LA implements AutoML only for the analysis aspect of SCA. An open challenge to using DL-LA is that there are no clear advantages to DL-LA for the significance of side-channel traces [26]. This challenge demonstrates that if a significant advantage to leakage assessment is gained by utilizing DL-LA, then the DL-SCA techniques can also be used with DL-LA.

The increase in popularity of SCA in security has led to developments in both attack vectors and defensive countermeasures. Starting from DPA [3], to implementing AutoML methods of deep learning with TVLA [27], the security aspect of SCA from a defensive and offensive standpoint have increased in scope from simple power analysis to implementations of machine learning, demonstrating a considerable growth of the research area. The idea of utilizing AutoML for DPA is a growing research area that has openings for finding research in time-series data as well as with signal processing. AutoML techniques are therefore well suited to further application in time-series related tasks for state of the art DPA and SCA.

### 2.2. Foundational Experiments

This experiment is conceptualized as an extension to prior control flow recovery experiments [1]. Power-based side-channel attacks were used to extract information by using properties from a FSM. In this foundational work, the original experiment made use

of a single classifier, the k-nearest neighbors (KNN) classifier with heterogeneous states for the FSM. KNN was able to achieve 81% or higher accuracy of transition classification. Accuracy would increase as the number of classes decreased. As a result, FSM components that handle sensitive information could be vulnerable to power-based side-channel attacks, even with only a single classifier being used to analyze the state machine.

The future work section in this foundational work [1] mentions how an avenue of research could be with how modifying the input to the states could be explored. In particular, the modifications to the states to also include homogeneous states as well as implementing tests using more than a single classifier were used as motivation for the extensions and further experimentation presented herein.

## 3. Methods

Analyzing the ability for program control flow to be recovered via power-based SCA required the creation and capture of a dataset encompassing numerous execution paths through a program while being able to associate the captured power traces with a training label for later machine-learning backed analysis of transition order. We explore both aspects of these experiments in two methodology sub-sections. The first details the adaptation of data collection from prior works [1] for the purposes of this experiment. The second details the training and evaluation of various machine learning models for the recovery of control flow information from the gathered trace data. Additionally, in the absence of a standard SCA benchmarking suite, and in the interests of reproducability, the entire code base is made available through a public GitLab repository [28]. Code made available in this manner is licensed under the GPLv3.

### 3.1. Data Generation

Data generation made use of the ChipWhisperer [2] family of devices. These are purpose-built microcontrollers for SCA data collection which contain all of the processing on-board for the collection of power traces. The ChipWhisperer Nano (CW Nano) is one such device backed by an STM32F0 microcontroller.

Each instance of the CW Nano device was programmed with a minimal C program which emulated a two-state state machine. See Figure 1 for a depiction of the FSM. The device was fed a transition sequence from a host device which communicated with the CW Nano before and after experimentation to send the oracle transition sequence and retrieve the power trace captured onboard the device during execution. The state machine transitioned through eight states in accordance with each bit of the oracle: a bit of 0 at position $i$ of the oracle indicates that the $i$-th state was state 1 while a 1 indicates the state was state 2. Both states performed integer addition a specific number of times where this number was determined during compilation and will be referred to as the value $w$ for the firmware. A firmware where $w = 1$ indicates that the firmware performed a single addition in each state while a firmware with $w = 16$ indicates that the states of that firmware performed the addition a total of 16 times.

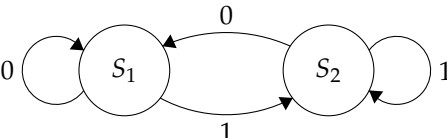

**Figure 1.** A diagram of the state machine executed by the target board. Transitions represent the next value in the oracle text. For instance, if the FSM was in state $S_1$ and the next digit was a "1", then the state machine would transition to state $S_2$ and execute the code associated with that state. If the FSM was in state $S_1$ and the next digit was a "0", then the state would transition to state $S_1$ and repeat the previously executed code.

Each CW Nano was paired with a Lenovo ThinkCentre running Ubuntu LTS 22.10. All collection used Python v3.10.6 and the ChipWhisperer library version 5.7.0 distributed

by PyPi. The CW Nano device was programmed with firmware version 5.1.0. All version numbers presented were the latest releases at the time of data collection. The code for handling the state transitions is presented in Figure 2 and an example of the state code for firmware ($w = 2$) is presented in Figure 3.

## State Transition Code

```c
for (uint8_t i = 0; i < 8; i++ ) {
    uint8_t state = transitions & 0x1;
    transitions >>= 1;
    if (state == 0 ) {
        worker(one_zero, zero_one, dest);
    } else {
        worker(one_zero, zero_one, dest);
    }
}
```

**Figure 2.** The C code for turning an oracle byte previously received by the CWNano into a series of state transitions on the device. The values `one_zero`, `zero_one`, and `dest` are discussed in Section 3.1.

## State Code

```c
void worker(int* x, int* y, int* total) {
    *total = *x + *y;
    *total = *x + *y;
}
```

**Figure 3.** The C code executed in the body of the state-machine. Figure 2 invokes this function with specially crafted arguments as discussed in Section 3.1. The number of times `*total = *x + *y;` is repeated is identified as $w$, so the code snippet above has a $w$ value of 2.

The inputs to the worker function of each state were specially crafted to ensure an equal hamming weight of all inputs to prevent the inference of the state transitions from the contents of the state by the machine learning models later trained. For further discussion of experiments which utilize differences in state behavior to enable related analysis, see [1]. The value of `one_zero` is 16 ones followed by 16 zeros, or 4,294,901,760 (base ten). The value of `zero_one` was the opposite, 16 zeros followed by 16 ones, or 65,535 (base ten). Thus, the total hamming weight of operands utilized across the body of the worker function is constant between states.

All potential oracle values, representing all 256 potential permutations of state transitions, were executed 100 times. Firmware was generated for all $w \in 1, 2, 4, 8, 16, 32, 64, 128$ and traces were captured across all oracles and 100 repeated executions and stored for later analysis. Labeling incorporated both the number of operations executed within each state of the firmware and the oracle used to generate the trace as well as the order in the 100 samples to uniquely identify each trace. The resulting data was 5.8 GB for each firmware, resulting in a total data set measuring approximately 40 GB in size.

### 3.2. Machine Learning Classification

For the task of recovering control flow information, we reduce the task to one of multi-class classification; this makes it a suitable task for applied machine learning classifiers. Each trace is labeled with the oracle byte used to dictate the state transitions, thus a proper classification would represent a complete recovery of the state transitions executed by the CW Nano device. As an example, consider a trace labeled with the oracle byte of $11001100_2 = 204_{10}$. The power-trace presented to the ML model would either be correctly classified as class 204, indicating a complete recovery of the control flow of

the program, or be incorrectly classified into another class, representing an inability to completely recover the control flow of the application.

Classification was completed by a number of classifiers provided by the Scikit-learn [29] (SKL) library (version 0.24.2 as packaged by conda forge). The selected classifiers had minimal configuration beyond the defaults provided by SKL, so further hyperparameter optimization may find ways to improve the models created by the process described herein.

Each training process was repeated across five folds of cross validation to address concerns of over-fitting. Eighty percent of the available data was used for training the classifier while another 20% was used for testing the complete classifier. Unless otherwise specified, all results presented in the rest of this work refer to metrics obtained by evaluating the testing dataset. The process was repeated in its entirety for each distinct firmware.

Data were taken directly from the dataset previously generated and split into cross-validation folds using a stratified k-fold method provided by SKL. No preprocessing was performed on the data. Four classifiers were then fit to the training data: a random forest classifier, a decision tree classifier, a KNN classifier, and a logistic regression classifier. The only configuration provided was to the logistic regression classifier; both a solver and maximum number of iterations were provided since without them, the process of fitting data caused convergence failure errors. Convergence failure errors emerged due to the fact that the provided number of iterations was insufficient to converge to a reasonable solution and the solver was needed to match the types of data generated by the CW Nano. Execution of these classification tasks was aided through parallel computation by placing the entire workflow for each distinct firmware on separate threads (i.e., $w = 1$ on one thread, $w = 2$ on another, etc.).

## 4. Results

For each firmware, with the exception of firmware where a single execution is performed ($w = 1$), classification accuracy values approaching 100% are observed for the random forest classifiers. Accuracy values of 98%+ are seen for decision trees and logistic regression. The KNN classifier is the outlier with an observed lower bound on accuracy values that was slightly greater than 80%.

Firmware with only a single execution of the addition operation ($w = 1$) was the exception to these metrics of accuracy. The resulting skew in overall performance is illustrated in Figure 4 while the exact performance of all four classifiers on each of the folds of testing is illustrated in Figure 5. The highest observed accuracy for this firmware ($w = 1$) was associated with a single fold of validation and the KNN classifier; it was only able to achieve a maximum accuracy of 3.26%. While this is nearly an order of magnitude more accurate than randomly guessing the class, it is far from a desirable accuracy. Preprocessing, ensemble classification, and hyperparameter optimization would be relevant approaches to addressing this concern if the classification of this firmware were the primary goal; however, as the goal is characterization of the bounds of potential classification, this is left for future work. Therefore, we can conclude that when the amount of work performed in each branch of control flow (i.e., when the amount of work performed by each state) is low, "off-the-shelf" machine learning models will struggle to determine the underlying program execution flow.

In stark contrast to the classification accuracy of traces obtained from this firmware ($w = 1$), as the $w$ value of the firmware increases, the performance capabilities of simple machine learning classifiers are well suited to the classification task set before them. For firmware with $w$ values of 2, 4, and 8, a random forest classifier was able to correctly recover the program execution flow in all five folds of cross-validation with 100% accuracy. For algorithm specific and cross-fold specific performances on firmware with two executions of addition operations ($w = 2$), see Figure 6. While all other classifiers achieved high levels of accuracy on average (99%+) over the same firmware, only the random forest classifier achieved this level of performance.

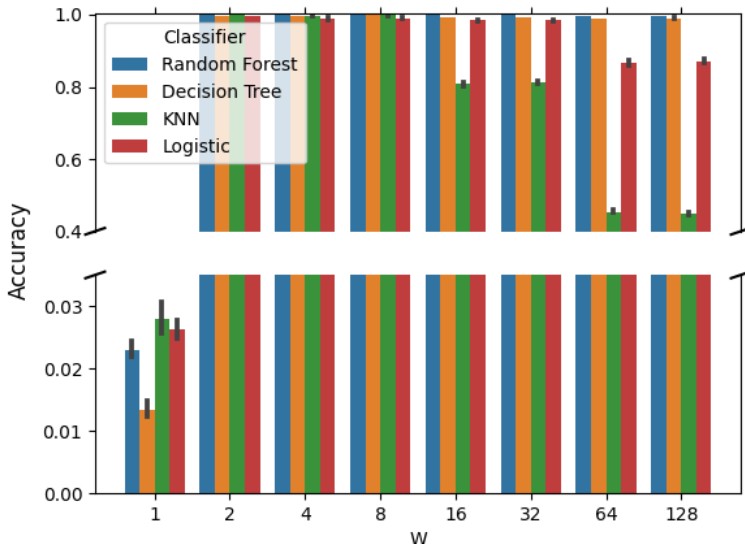

**Figure 4.** A figure which demonstrates the variance, or lack of variance, exhibited by each classification algorithm depending upon the number of operations performed by the firmware. The y-axis is split to emphasize the difference between $w = 1$ and the other $w \geq 2$.

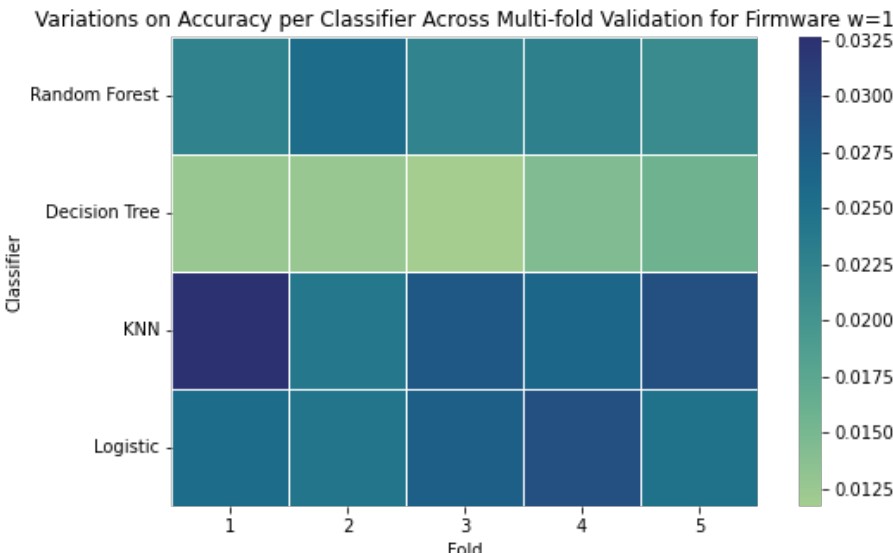

**Figure 5.** A heatmap demonstrating the accuracy of the testing phase for firmware with a single operation in each state ($w = 1$). Most notable is that, while results are better than random guessing, classification accuracy of 2% is extremely different than the 98%+ accuracy achieved for all other tested firmware.

Across all classifiers, a decrease in overall accuracy was observed moving from firmware with 8 executions ($w = 8$) to firmware with 16 ($w = 16$). In the case of random forest classifiers, this is a decrease from an average accuracy of 100% to 99.92%. Both the decision tree classifiers and logistic regression had their accuracy decrease as the number of operations performed by the firmware increased, but both remained over 98% classification accuracy on average. The KNN classifier saw vastly diminishing performance in the move to firmware with more operations, only achieving a maximum accuracy of 81.4% with an average accuracy of 80.8%. Similar decreases in performance were observed when moving to the next level of firmware ($w = 32$). A visual presentation of the average performance across these various firmware with more than two operations is available in Figure 7.

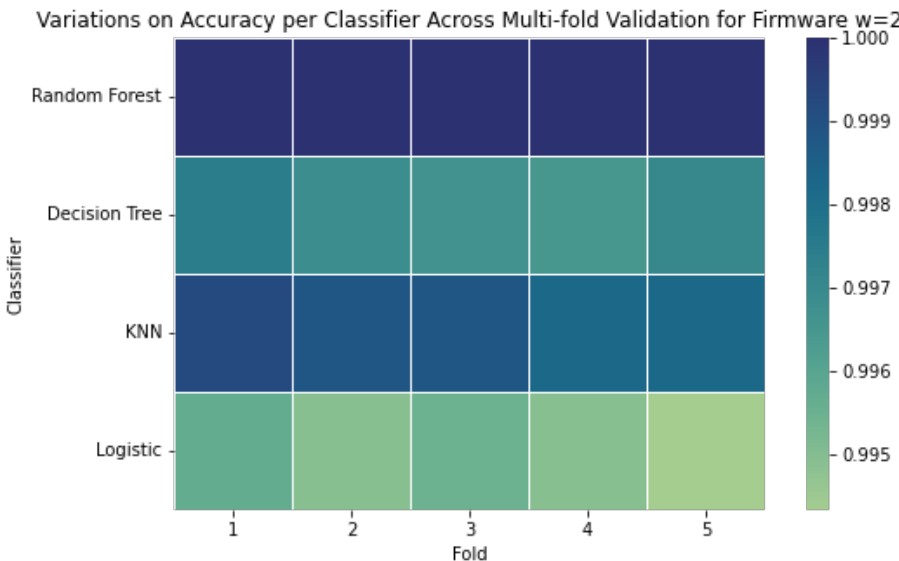

**Figure 6.** A heatmap demonstrating the accuracy of the testing phase for firmware with two operations executed in each state ($w = 2$). This firmware exhibits a similar difference between each classifier as seen in Figure 5 but shifted to the high end of classification accuracy. The random forest classifier maxes out at 100% accuracy for this firmware.

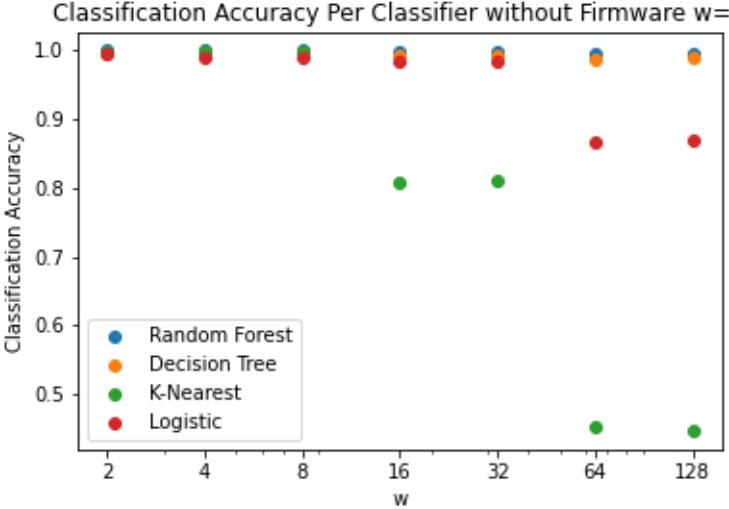

**Figure 7.** A series of dot plots which illustrate the performance of various classifiers for firmware with various amounts of computation. Accuracy for firmware of $w = 1$ is not included since it was less than 3% for all four tested classifiers. See Figure 5 for specific performance on firmware with $w = 1$ for each classification algorithm.

Further hyperparameter optimization might be effective in improving accuracy in firmware where more operations are performed in each state of the state machine, and automated approaches to the machine learning for this task could find effective preprocessing and postprocessing to improve the overall results. However, as a proof of concept and demonstration that the work performed in each branch of the state machine is not required to be different in order to recover the transitions, these results are highly significant.

## 5. Discussion

A number of conclusions can be drawn from the results achieved across these classification problems. Beyond the minimal proof of concept that control flow can be significantly recovered even when the work in different states of a limited purpose finite-state machine

are identical, these findings suggest that obfuscation techniques may be necessary to obscure the control flow of the program when said control flow conveys security-relevant information. As an example, an attacker should not be able to purchase a smart lock, determine the device's control flow from the embedded finite-state machine, and then gain the ability to access a home using the same smart lock model. Since the results show that this information was exfiltrated from the underlying micro-controller, this is a potential vulnerability that may require more sophisticated protections than traditional, software-based ones.

One surprising result was the extremely low accuracy while applying machine learning classifiers to the firmware which has only one operation ($w = 1$). It was hypothesized before data analysis was completed that the average accuracy would decrease monotonically as the number of operations increased. The intuition used to develop this hypothesis was that as the proportional time spent in the control flow code sections increased, the more accurate the transition recovery would be. This followed from the observation that, when more relative time is spent in control flow, more of the data points captured within the power trace would be directly related to the process of determining state machine transitions. The subversion of this hypothesized outcome indicates that the number of operations has much less influence in the classification of transitions than it was initially assumed. It is clear that the amount of work performed in a state still has influence, as evidenced by the variation in classification accuracy in correlation with the number of operations within each state. However, this role may not be nearly as important as the actual work performed and the state transitions executed by a low-powered device.

The first research question sought to identify whether different behavior is required in each state of an embedded state machine in order to completely recover the transitions. This question is answered firmly in the negative. While prior works made use of the different behavior of a heterogeneous, two-state FSM to more easily perform the classification [1], this difference is not required. This work clearly demonstrates that, while differences in state behavior can allow for recoverable transition sequences, it is not strictly necessary since high transition recovery accuracy was achieved with homogeneous state behavior. Even when applying the same techniques and classifiers as were used in prior work, meaningful levels of accuracy were achieved with said states.

This finding suggests that power-based side-channel attacks will be an applicable tool to recover state transition information regardless of what kind of work is performed by the states within an FSM. For devices with minimal numbers of states which do a meaningful amount of work (i.e., they are not comparable to the $w = 1$ firmware tested in these experiments), these results suggest it is possible to recover the transition order of the underlying FSM. As of the time of writing, such behavior is also consistent with speculation in current literature. Further work will be necessary to determine if these findings are consistent when expanded to state machines with many states.

The second research question sought to address to what degree the proportion of time spent in control flow and in a particular state of an FSM would impact the recoverability of the state transition ordering. It was initially hypothesized that the ratio of time spent in the state machine versus in the control flow would be the primary predictor of classification accuracy. While this held true for firmware with more than one operation ($w \geq 2$) from 2 to 128, the special case was the firmware with a single operation ($w = 1$). It can therefore be concluded that there is a bounded range in which the control flow can be recovered without more advanced means than are presented here. Further research will be necessary to determine whether the lower bound is one which can be encountered in production grade FSM. However, the lower bound of recoverability suggests that minimized states may evade detection, recovery, and classification. This is demonstrated by the low accuracy associated with classifying for firmware with a single operation.

The final research question examined how effective "off-the-shelf" machine learning models are when applied to the task of recovering control flow information. "Off-the-shelf" machine learning models, specifically more light-weight ones than the deep learning

approaches common in power-based side-channel attacks, show much promise in their ability to capture high-level control flow information. Even without hyperparameter optimization, high levels of accuracy were achieved. With this optimization, it may be possible to achieve similar accuracy on firmware with more operations. Situations in which "off-the-shelf" solutions are not sufficient to achieve high classification accuracy must be further explored to motivate the need for hyperparameter optimization.

The efficacy of random forest algorithms for classification tasks is well known. Yet even with this reputation, these algorithms' performance on the collected data is important as it may suggest their applicability and strong performance on more complex state machines. When paired with optimization algorithms such as Bayesian Optimization, their capabilities may continue to be relevant, potentially minimizing the need for more computationally challenging solutions which make use of deep learning such as DL-SCA.

## 6. Conclusions

In this work, the program control flow was able to be recovered using SPA. The classification models used were able to achieve a high level of accuracy, with the random forest model reaching 100% accuracy for three of the values of $w$ tested; the other models also reached very high classification accuracy, with several averaging over 98% on firmware operations with more than one execution ($w \geq 2$). This level of performance was achieved without hyperparameter optimization begin applied, an approach which could lead to improvements in some situations.

Power-based side-channel analysis is a new potential tool for attackers looking to recover control flow information from an embedded or otherwise low-powered system. Due to the recent nature of the development of meaningful recovery attacks on program control flow information, it remains to be seen to what degree existing counter-measures will be applicable to the protection of this information. Contrary to the intuition of this research team, short sections of code were able to evade detection with meaningful impacts on the accuracy of recovery as exhibited by the results surrounding firmware with a single execution of operations in this work.

Looking to the future, most current work in power-based control flow recovery has been done with state machines which have only two states; however, in practice, nearly all state machines have many more states. Future work should explore the potential for the current two-state techniques to be extended to multiple homogeneous states as well as multiple heterogeneous states and variations between the two. Furthermore, the applicability of automated machine learning pipelines for the task of classifying state transitions is another avenue for future exploration. While current "off-the-shelf" machine learning classifiers are sufficient to classify the states under consideration, the ability of these findings to be transferred to more complex state machines must be examined to discern where their application breaks down and traditional AutoML pipelines must be employed. Overall, with the ability for FSM transitions to be recovered clearly established now, the task must turn to more firmly defining the capabilities and bounds on this avenue of attack. Variable transition counts, wildly different state computations, more complex state machines, and further perturbations on the environments data is collected within are all promising future directions of research at this time. This work should seek to qualify the limitations of SPA approaches for FSM transition recovery through power analysis. Finally, historic mitigation techniques applied to combat side-channel attacks must be re-evaluated to determine to what extent existing mitigations protect against this control flow recovery attack.

One primary limitation is that the comparative time spent in each state, a by-product of the number of operations in each state, is constant across a sample. While the question of whether a simple machine learning model would be sufficient for recovering the control flow of a program with identical work in each state has been answered in the affirmative in this paper, future work should certainly address this limitation by determining if a consistent time in each state is necessary for these results to be widely transferable. Perhaps more

specialized machine learning may be able to improve the classification, but in comparison to the other models created throughout this process, it is clear that the data captured itself is the limitation directly responsible for reducing general classifier accuracy.

Additionally, while attempting to make use of "off-the-shelf" machine learning models was a key research question, it does assume that the benefits of deep learning approaches, or other AutoML approaches such as NAS, are not of enough importance to justify the computational trade-offs. The strong performance of the algorithms examined in this work may justify this restriction in this specific scenario, but further examination of their limitations will be necessary to determine if more modern, advanced, or complex machine learning pipelines allow for more meaningful state transition recovery in more complex applications.

**Author Contributions:** Conceptualization, A.R. and C.C.; Methodology, A.R.; Software, A.R.; Validation, S.O.; Investigation, A.R. and S.O.; Resources, M.B.; Data curation, A.R.; Writing—original draft, A.R., S.O., J.B., C.C. and M.B.; Writing—review & editing, A.R., S.O., J.B., C.C. and M.B.; Visualization, A.R., S.O. and J.B.; Project administration, M.B.; Funding acquisition, M.B. All authors have read and agreed to the published version of the manuscript.

**Funding:** This work was supported through various contracts and gifts including INL Laboratory Directed Research & Development (LDRD) Program under the DOE Battelle Energy Alliance Standard Research Contract #249922, IOG, and the University of Wyoming's Nell Templeton Endowment.

**Data Availability Statement:** The data presented in this study are openly available in FigShare at 10.6084/m9.figshare.23635623.

**Acknowledgments:** The research team would like to acknowledge and thank the Secure Systems Collaborative for their assistance in revisions and presentation of information.

**Conflicts of Interest:** Any opinions, findings, and conclusions or recommendations expressed in this material are those of the author(s) and do not necessarily reflect the views of any agency, sponsor, or corporate entity.

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
