# Peer review of "Power-Based Side-Channel Attacks on Program Control Flow with Machine Learning Models"

_jcp, doi:10.3390/jcp3030018_

Round 1

Reviewer 1 Report

This paper is interesting and applies more advanced machine learning techniques to state machines which perform identical operations in all branches of control flow. However, I have some comments as follows:

1. Title of this paper should be modified to address machine learning algorithms.

2. Authors said that they used advanced machine learning techniques. where is it? It is basic algorithms for machine learning.

3. Introduction has to rephrase. Authors have to remove subsections heading only. 1.1. Motivation and 1.2. Contributions.

4. 2. Prior Works should be renamed to Related Works.

5. Conclusion has to rephrase. Authors have to remove subsections heading only. 6.1. Future Work and 6.2. Limitations.

6. Check the format of the paper including the references and make the proper corrections where are necessary.

Despite these shortcomings, this paper is considered to be accepted with minor revision.

Reviewer 2 Report

Why the research in the paper is important:

The control flow of a program represents valuable and sensitive information; in embedded systems, this information can take on even greater value as the resources, control flow, and execution of the system have more constraints and functional implications than modern desktop environments. Early works have demonstrated the possibility of recovering such control flow through power-based side-channel attacks in tightly constrained environments; however, they relied on meaningful differences in computational states or data dependency to distinguish between states in a state machine.

Contributions and novelty of the paper with respect to the state of the art:

This work applies more advanced machine learning techniques to state machines which perform identical operations in all branches of control flow. Complete control flow is recovered with 99% accuracy even in situations where 97% of work is outside of the control flow structures. This work demonstrates the efficacy of these approaches for recovering control flow information; continues developing available knowledge about power-based attacks on program control flow; and examines the applicability of multiple standard machine learning models to the problem of classification over power-based side-channel information.

See my comments regarding PQC and SCA and lightweight crypto detailed below.

- With any new security measure implementation, you need to make sure you provide benchmark for active/passive side-channel attacks (SCAs). Mention and add a paper about "fault detection of ring-LWE on FPGA" too.

- With the advent of post-quantum cryptography (PQC), it is better to add some relevant works to make sure you cover that topic too. This is the hottest topic in cryptography now. When PQC replaces ECC/RSA every security application from smart phones to block chains will be affected. With PQC, add a previous paper on each of these separately: (a) Curve448 and Ed448 on Cortex-M4, (b) SIKE on Cortex-M4, (c) SIKE Round 3 on ARM Cortex-M4, (d) Kyber on 64-Bit ARM Cortex-A, (e) Cryptographic accelerators on Ed25519.

- NIST lightweight standardization was finalized in Feb. 2023. Also mention fault attacks as side-channel attacks, these topics to explore and add a paper reference on each of these separately: (a) Fault detection of architectures of Pomaranch cipher, (b) reliable architectures of grostl hash, (c) fault diagnosis of low-energy Midori cipher, (d) fault diagnosis of RECTANGLE cipher.

- References are not uniformly formatted.

- Please add a subsection and one or more future works for enhancing your presentation

- DPA+DFA for example can be mounted at the same time and their combined countermeasures (for example TI and Error Detection Schemes) can be used for thwarting attacks in combined manner (need to be discussed). Adding a paragraph suffices.

It is readable.

Round 2

Reviewer 2 Report

Paper can be accepted.